# Administrative Division Data of Grand Casablanca: Creation of a District Repository Using QGIS

**Khaoula Ajbal** [1,2,*] , **Samy Housbane** [1,2] **and Bennani Othmani Mohammed** [1,2]

1    Medical Informatics Laboratory (LIM), Medicine and Pharmacy Faculty of Casablanca, Hassan II University, Casablanca 20000, Morocco; s.housbane@gmail.com (S.H.); bennanim2000@gmail.com (B.O.M.)
2    Neurosciences and Mental Health Laboratory, Medicine and Pharmacy Faculty of Casablanca, Hassan II University, Casablanca 20000, Morocco
*    Correspondence: ajbal.khaoula@gmail.com

**Abstract:** In the context of a public mental health study conducted in Casablanca, the economic capital of Morocco, we encountered a serious challenge regarding the availability of consistent and accurate administrative division data. Hence, using the Google Maps API in QGIS, we proceeded to geocode the Grand Casablanca districts and overlay them on the updated Morocco's administrative regions shapefile. The district data were summarized and gathered creating an administrative division repository from the district level up. Thus, the main contribution of the current paper, is a table containing (District, District_CP, Commune, Type_com, Prefecture/Province (PR), PR_CP, Region) and the second output is the shapefile that generated the table. The GIS data and the map are useful to other researchers in Morocco and elsewhere who have had no opportunity to access administrative division data of the location. As for foreign researchers from other developing countries the paper's approach can be applied in their studies to create other lacking geographic repositories.

**Keywords:** QGIS; Geocoding; Casablanca administrative division; shapefile; district repository

---

## 1. Summary

The lack of information and data resources in developing countries represent a major drawback for the research field. The administrative division data of Morocco, even if available on the net [1] in the form of a shapefile, created and last updated in October 2016, lacks consistency as it was based on the 2009 administrative division of Morocco [2]. It also lakes a certain level of details needed for the creation of a district repository. Taking for instance one of the largest regions of Morocco, Grand Casablanca, where the economic capital of the country is located, still, there is no data on the official website of its territorial administrative division [3], neither a detailed table nor an editable map.

Hence, at first we had to collect all Grand Casablanca district names along with their respective ZIP codes [4] from the official Moroccan ZIP code website. And then, after updating the Moroccan administrative division shapefile by reassigning each commune to its prefecture or province, according to the 2015 general monograph of the Region Grand Casablanca [5]; we finally were able, using QGIS (Quantum Geographic Information System), to geocode the gathered districts over the updated shapefile and extract the data creating the Grand Casablanca District repository. The latter was used in two separated under review articles by the same current authors. One titled "Preprocessing unstructured geographical data for public health analytics applications" and the other "Patient profiling

for mental healthcare management optimization in developing countries using Datamining". The GIS (Geographic Information System) data and the map are useful to other researchers in Morocco and elsewhere who have had no opportunity to access spatial data of the location. As for foreign researchers from other developing countries the paper's approach can be applied in their studies to create other lacking geographic repositories. The data can also be used by city practitioners looking for up to date detailed administrative division datasets.

## 2. Data Description

The raw data is gathered in one table GC-repository (Grand Casablanca repository) containing 402 records of the following fields detailed in the table below (Table 1): District, District_CP, Commune, Type_com, Prefecture/Province, PR_CP, Region. The second output is a shapefile that generated the table, built by geocoding the districts and overlaying them on a larger map, that has been updated to the latest administrative division. The description of the shapefiles and their content is detailed in the methods section.

**Table 1.** Data table's attributes description.

| Attribute | Type | Description |
|---|---|---|
| District | String | District name |
| District_CP | Number | District ZIP code |
| Commune | String | Commune name (lowest territorial division level gathering districts) |
| Type_com | String | Commune type depending on the urban or rural predominance |
| Prefecture/Province | String | Prefecture or Province name (contains communes) |
| PR_CP | Number | Prefecture or Province ZIP code |
| Region | String | Region name (highest territorial division level) |

## 3. Methods

The Software used for the creation of the data is QGIS 2.18.24. The main idea was to geocode districts' names using a Google maps API extension and overlay them on the administrative division map of Morocco (the shapefile can be downloaded from here [4]). We will be explaining in details the procedure in the following steps along with the screenshots:

### 3.1. Creation of District_CP.csv

At first we gathered all the district names along with their respective ZIP codes according to the new administrative division of Grand Casablanca [4] from the official Moroccan ZIP code website. Then, we saved the data in a CSV file District_CP.csv formatted for UTF-8.

### 3.2. Installation of MMQGIS Plugin in QGIS for Google Maps Geocoding

We chose to geocode with Google Maps API, using a free key provided by Google's Get API. Then we loaded the QGIS plugin repository by selecting "Plugins –> Manage and Install Plugins" from the menu bar. To replicate this, one may see the window shown in (Figure 1) containing all the plugins and their settings, select and install MMQGIS by checking the box next to it, and then click on the install plugin button.

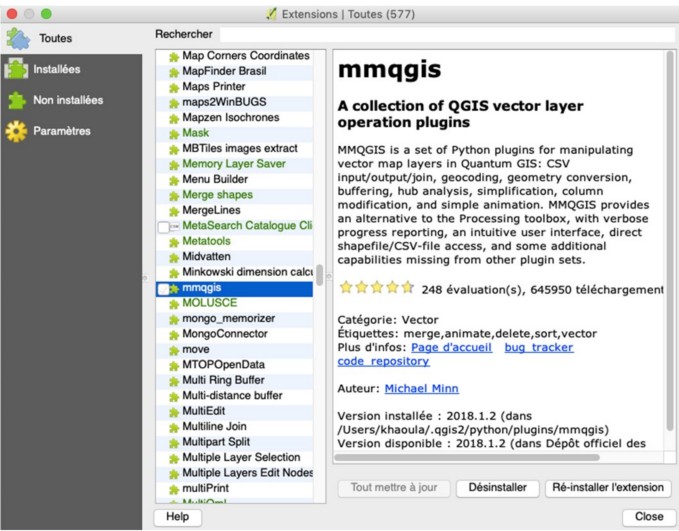

**Figure 1.** Install MMQGIS Plugin.

### 3.3. Creation of Region Grand Casablanca Using MAR_adm Shapefiles

The new administrative division of Grand Casablanca integrated the provinces of El Jadida and Sidi Bennour of the former region of Doukkala-Abda and the provinces of Settat, Benslimane and Berrechid of the former region of Chaouia-Ouardigha in the new region of Casablanca-Settat [4]. Hence after downloading the shapefiles MAR_adm of the Moroccan administrative regions, from the VGI OpenAfrica [1], we used the query builder at the bottom of the layer properties window (Figure 2) to select the wanted prefectures and provinces from the shapefile MAR_adm3:

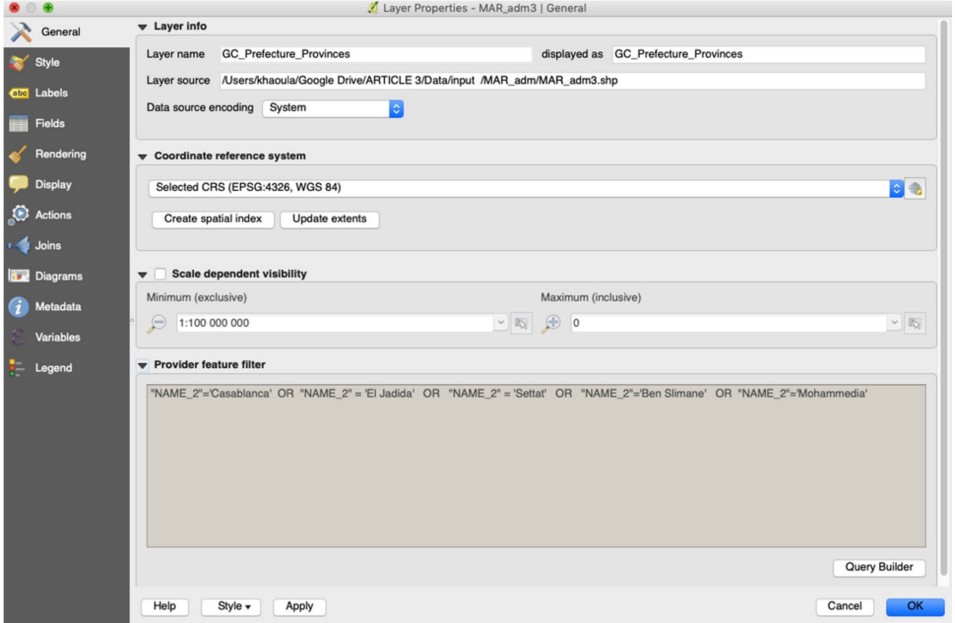

**Figure 2.** Creation of Grand Casablanca region.

"NAME_2" = 'Casablanca' OR "NAME_2" = 'El Jadida' OR "NAME_2" = 'Settat' OR "NAME_2" = 'Ben Slimane' OR "NAME_2" = 'Mohammedia'

Then to divide the prefectures and provinces of Grand Casablanca into communes we select from the menu bar Vector –> Geoprocessing tools –> Clip, which will open a window (Figure 3) where we specify the fields as follow:

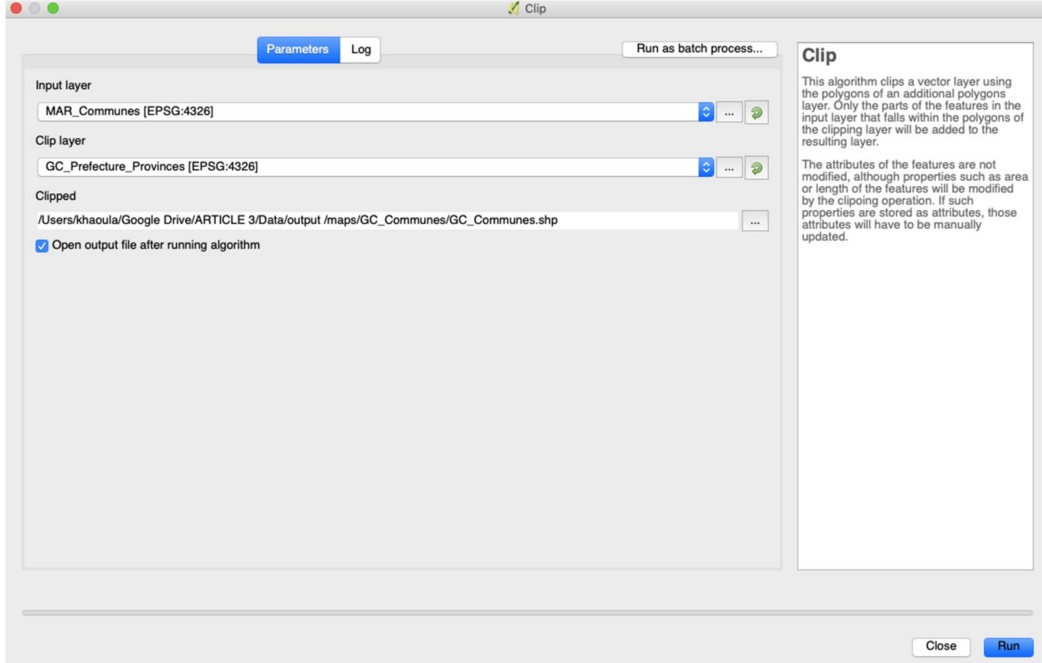

**Figure 3.** Division of Grand Casablanca into communes.

*3.4. Update of the Administrative Distribution of Communes Over to the Prefectures and Provinces*

Once we had extracted the division of Grand Casablanca into communes, we corrected the administrative repartition of communes over the prefectures and provinces (Figures 4 and 5) according to the 2015 general monograph of the Region Grand Casablanca [5], to be conformed to the map below Figure 6.

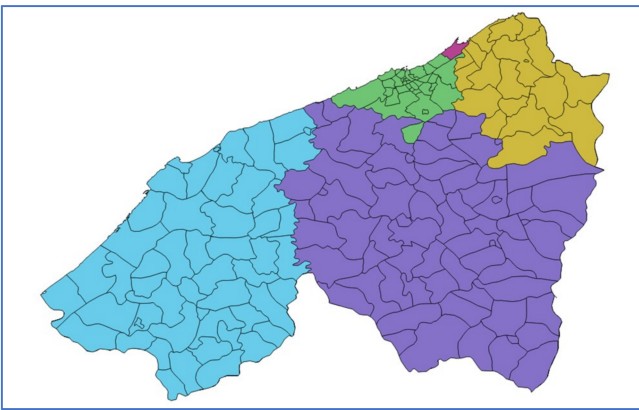

**Figure 4.** Before correction.

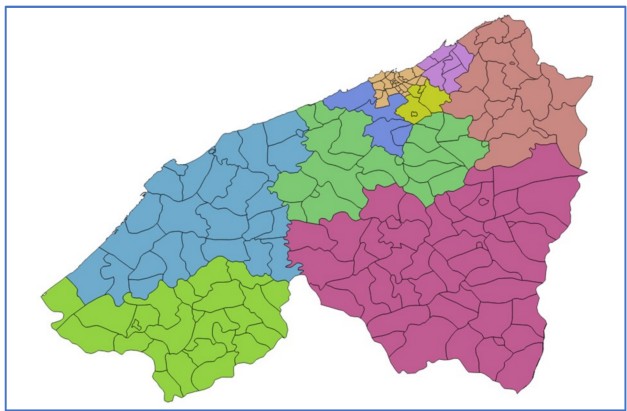

**Figure 5.** After correction.

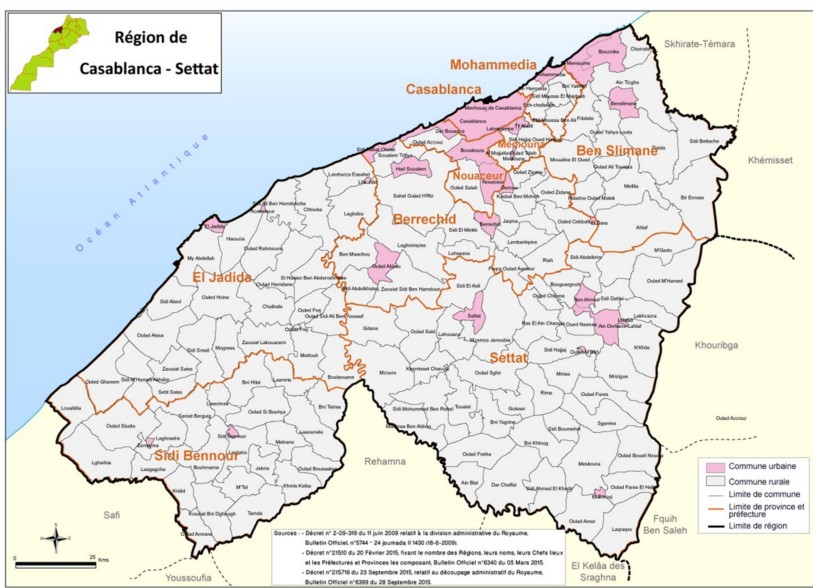

**Figure 6.** 2015's administrative division of Grand Casablanca.

*3.5. Geocoding the District Names*

Using the District_CP.csv file with utf-8 encoding created before and the Google maps API key, we geocoded the district names as addresses as shown below in the Figure 7, it takes a few minutes depending on the internet speed.

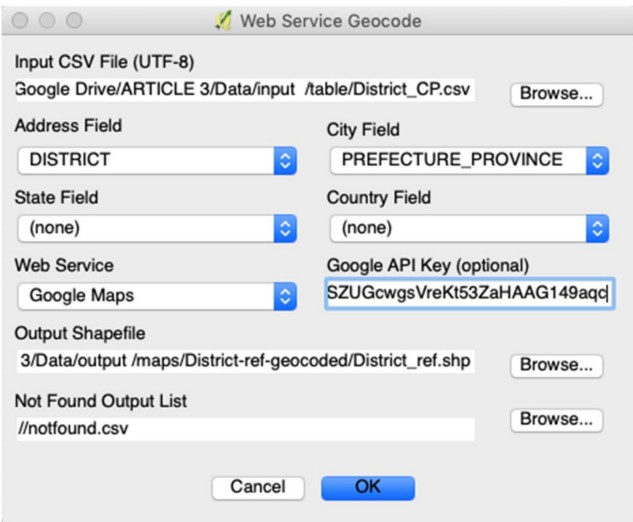

**Figure 7.** District geocoding using Google Maps.

*3.6. Consolidating the Data in One Table Containing District, Commune, Commune Type, Prefecture or Province, Region and Their Respective Postal Codes*

MMQGIS uses Google maps as a base map, and it is advised to use a Google Earth background map to ensure the good quality and precision of geocoding. In our case, since we have used our own shapefile as a background map, some districts were misplaced. Obvious misplacements, like the ones outside the land, were corrected automatically by dragging them into the nearest part of land. Other less noticeable ones, especially the ones over some communes' boundaries, were handled individually by looking for their right commune, in some cases referring to Wikipedia, in others, to our knowledge of the area. After correcting the geocoding misplacements and adding a few missing ones using the Toggle editing on the generated layer District-ref.shp, we proceed to consolidate the data with the revised commune layer GC_Communes_revised.shp by selecting from the menu bar Vector –Data management tools –> Join attributes by location (Figure 8).

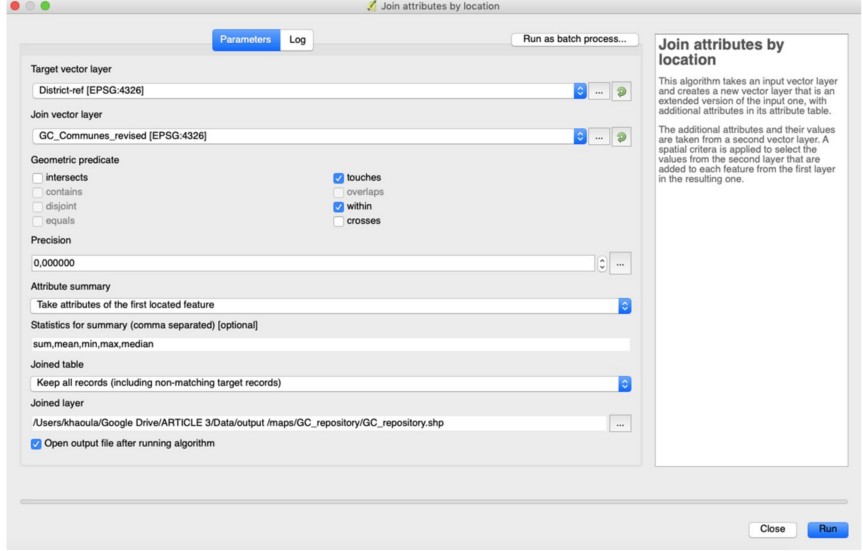

**Figure 8.** Data consolidation.

Then we entered the names of the layers to be joined, specified the geometric predicate as 'touches' and 'within', selected in joined table 'keep all records' and after specifying the name of the target shapefile GC_repository.shp, we clicked the button Run.

The generated shapefile can be used as a base map that can be easily edited to fit any administrative related needs covering the management of different areas such as human resources, medical facilities, finance or support services.

*3.7. Data Extraction and Cleaning*

Each shapefile added in the layers panel contains its own data gathered in the form of a table that can be displayed by selecting Open attributes table after a right click on the layer.

Hence for the creation of the Grand Casablanca district repository, all that was left to do was to extract that table from the final layer containing the distribution of the districts on the updated administrative division layer. The extraction is done by saving the layer GC_repository as a CSV file with the same name after a right click on it.

We also cleaned up the data in the final CSV file, not the shapefile, by removing from the table some redundant columns caused by using different layers of the same region and other unnecessary fields for this case, however they can be easily recovered by repeating the last extraction maneuver.

## 4. User Notes

A read me note text file is attached to the data in the dataset link, it contains brief descriptions of the other files.

An additional folder of the output maps in GeoJson format is available on this link.

**Author Contributions:** Conceptualization, K.A. and S.H.; methodology, K.A.; software, K.A.; validation, B.O.M. and S.H.; writing—original draft preparation, K.A.; writing—review and editing, S.H.; supervision, B.O.M.; project administration, B.O.M.

**Funding:** This research received no external funding.

**Conflicts of Interest:** The authors declare no conflict of interest.

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
