# Peer review of "Administrative Division Data of Grand Casablanca: Creation of a District Repository Using QGIS"

_data, 2019_

Round 1

Reviewer 1 Report

Dear authors,

Follow are my comments for your manuscript. I’m starting with general comment and follow with comment per line number:

Acronyms as QGIS (Line 36), GIS (Line 41), GC-repository (Line 47), 205 Ko (Line 49) and MMGGIS (line 64) are not define

Lines 32-35: you wrote a confusing sentence: “And then, after updating the Moroccan administrative division shapefile to be conformed (follow) to the administrative repartition (re-divide) of communes (communities) over the prefectures (regions) and provinces according to the 2015 general monograph of the Region Grand Casablanca.” Is it mean that you took the 2015 shape file and adjust all other files (boundaries, names, geocoding) to that file? Please re-write this sentence. 

Line 49 – you wrote: “The second output is …..”. What is the first one? If it the table, what is the difference between the table and the shape file? If there is a difference, meaning the table itself is important even without the location of each place, can you explain how the table itself can be use?

Table 1 description of Commune: “lowest territorial division gathering districts” –you need to add the word “level” so it will be “lowest territorial level….”

Line 106 – you describe how you corrected geocoding misplacements. Please add example how did you find these misplacements for any reader future work. This will explain the need of quality-assessment for the original files

Author Response

Point 1: Acronyms as QGIS (Line 36), GIS (Line 41), GC-repository (Line 47), 205 Ko (Line 49) and MMGGIS (line 64) are not define

Response 1:

QGIS (Line 36): Quantum Geographic Information System (added)

GIS (Line 41): Geographic Information System (added)

GC-repository (Line 47): name of Grand Casablanca repository (added)

205 Ko (Line 49): was referring to the size of the file by Kilo-octets, but we have deleted it since it was irrelevant. (deleted)

MMQGIS (line 64): Michael Minn Quantum Geographic Information System plugin, it’s named after its developer Michael Minn.

Point 2: Lines 32-35: you wrote a confusing sentence: “And then, after updating the Moroccan administrative division shapefile to be conformed (follow) to the administrative repartition (re-divide) of communes (communities) over the prefectures (regions) and provinces according to the 2015 general monograph of the Region Grand Casablanca.” Is it mean that you took the 2015 shape file and adjust all other files (boundaries, names, geocoding) to that file? Please re-write this sentence. 

Response 2: And then, after updating the Moroccan administrative division shapefile by reassigning each commune to its prefecture or province, according to the 2015 general monograph of the Region Grand Casablanca…” (done)

It has been detailed in Methods section 1.4.

Point 3: Line 49 – you wrote: “The second output is …..”. What is the first one? If it the table, what is the difference between the table and the shape file? If there is a difference, meaning the table itself is important even without the location of each place, can you explain how the table itself can be use?

Response 3: Indeed, the table itself is an output as it has been used in both studies mentioned in the summary as a district repository used for natural language processing applied to an address field containing (district , commune , prefecture/province…). It helped structuring the address field for public health analytics applications.  

Point 4: Table 1 description of Commune: “lowest territorial division gathering districts” –you need to add the word “level” so it will be “lowest territorial level….”

Response 4: Commune name (lowest territorial division level gathering districts) (done)

Region name (highest territorial division level) (done)

Point 5: Line 106 – you describe how you corrected geocoding misplacements. Please add example how did you find these misplacements for any reader future work. This will explain the need of quality-assessment for the original files

Response 5: I have added this explanation to the section about the misplacements “MMQGIS uses Google maps as a base map, and it is advised to use a Google Earth background map to insure the good quality and precision of geocoding. In our case, since we have used our own shapefile as a background map, some districts were misplaced. Obvious misplacements, like the ones outside the land, were corrected automatically by dragging them into the nearest part of land. Other less noticeable ones, especially the ones over some communes’ boundaries, were handled individually by looking for their right commune, in some cases referring to Wikipedia, in others, to our knowledge of the area.

Reviewer 2 Report

Dataset is well explained. However, I have some comments:

Line 30 - comma is misplaced

Line 33-37 two sentences should be one. Please rewrite.

Line 38- mentioned papers should be referenced.

Line 47-sentence started with small letter 

Line 49 - 205 Ko?

Figures should be explained in the text.

English sshould be improved and whole paper checked for syntax errors.

Explain in one paragraph whole metodology you used from start to the end and then continue to detail explanations.

Describe problems you encountered during the process

Author Response

Point 1: Line 30 - comma is misplaced

Response 1: Taking for instance one of the largest regions of Morocco, Grand Casablanca, where the economical capital of the country is located, still, there’s no data on the official website…

Point 2: Line 33-37 two sentences should be one. Please rewrite.

Response 2: And then, after updating the Moroccan administrative division shapefile to be conformed to the administrative repartition of communes over the prefectures and provinces according to the 2015 general monograph of the Region Grand Casablanca [4]; We finally were able, using QGIS, to geocode the gathered districts over the updated shapefile and extract the data creating the Grand Casablanca District repository.

Point 3: Line 38- mentioned papers should be referenced.

Response 3: The papers mentioned are still under review, so I changed “submitted” with “under review” to be more specific.

Point 4: Line 47-sentence started with small letter 

Response 4: The raw data is gathered in one table GC-repository containing

Point 5: Line 49 - 205 Ko?

Response 5: It was referring to the size of the map file 205 Kilo-octets. On second thought it wasn’t relevant so I deleted it.

Point 6: Figures should be explained in the text.

Response 6: I have mentioned the figures in the text as requested.

Point 7: English should be improved and whole paper checked for syntax errors.

Response 7: It has been reviewed as requested and changes has been made.

Point 8: Explain in one paragraph whole methodology you used from start to the end and then continue to detail explanations.

Response 8: The methodology was explained in the summary from start to the end:

“Hence, at first we had to collect all Grand Casablanca district names along with their respective postal codes [3] from the official Moroccan postal code website. And then, after updating the Moroccan administrative division shapefile to be conformed to the administrative repartition of communes over the prefectures and provinces according to the 2015 general monograph of the Region Grand Casablanca [4]; We finally were able, using QGIS, to geocode the gathered districts over the updated shapefile and extract the data creating the Grand Casablanca District repository.” In the Method section, to avoid repetition, I have only cited the main idea: “The main idea is to geocode district names using a Google maps API extension and overlay them on the administrative division map of Morocco (the shapefile can be downloaded from here [4]). We will be explaining in details the procedure in the following steps…”. Unless I should exchange them?

Point 9: Describe problems you encountered during the process

Response 9: The first problem encountered was while looking for the Moroccan administrative division shapefile, that we finally found but had to update.

…we encountered a serious challenge regarding the availability of consistent and accurate administrative division data..

The second was about insuring the quality of the data generated after the geocoding, since it is advised to apply it over a Google earth background map. However, in our case we have used one available on the net, therefore lacking consistency.

MMQGIS uses Google maps as a base map, and it is advised to use a Google Earth background map to insure the good quality and precision of geocoding. In our case, since we have used our own shapefile as a background map, some districts were misplaced. Obvious misplacements, like the ones outside the land, were corrected automatically by dragging them into the nearest part of land. Other less noticeable ones, especially the ones over some communes’ boundaries, were handled individually by looking for their right commune, in some cases referring to Wikipedia, in others, to our knowledge of the area.” (added)

Reviewer 3 Report

    Data description:

This dataset is an update of original data from Morocco, following a new administrative repartition of communes (2015).
The date of the original dataset should be provided.

    Methodologies:
There is no direct data collection, intstead it is the combination of a dataset with a more recent administrative districting. The tools for realizing that combination are Google Maps and QGIS plugins. Though I didn't try to reproduce the described process, the given details for using these tools seems to be sufficient.

    Metadata:
This is the weakest part of this paper. Some more work should be spent detailing and completing the metadata.
In particular, different fields must be dated: we understand that the new districting has been issued in 2015, but how old is the previous one?
Interestingly, both districtings could be saved in the same dataset, allowing backward compatibility.

    Copyright license, Archives, Reuse:
open CC0, with doi number, reusable.

    Data quality:
The authors declare having made a few corrections, without further describing the origin of these errors. A few more comments would be welcomed. The guess is that the new dataset shares the same level of quality than the previous one, and broken down into the same types of quality, excepting the time dimension.

    Format:
the proposed formats are ok (CSV and ShapeFile). Would it be possible to provide a GeoJson as well?

    CONCLUSION

My bet is that the modifications asked above (eg. metadata and quality) are rather minor. However, they must be present in the final version of the text, before any publication approval.

Author Response

 Point 1: This dataset is an update of original data from Morocco, following a new administrative repartition of communes (2015).

The date of the original dataset should be provided.

Response 1: The original dataset was created and last updated in October 25th 2016. Even if the creation date is more recent, it was based the old administrative repartition, perhaps for its unavailability by then.

“The administrative division data of Morocco, even if available on the net [1] in the form of a shapefile, created and last updated in October 2016, lacks consistency as it was based on the 2009 administrative division of Morocco(added)

Point 3: This is the weakest part of this paper. Some more work should be spent detailing and completing the metadata.
In particular, different fields must be dated: we understand that the new districting has been issued in 2015, but how old is the previous one?
Interestingly, both districtings could be saved in the same dataset, allowing backward compatibility.

 Response 3: The update that has been made, was on the communes’ level, as for each one of the 201 communes (Name_3) we updated the prefecture/province field (Name_2) to be conformed to the new 2015 division. In other words, the previous Name_2 field was overwritten. However, both the old and revised communes’ shapefiles are available in the output folder.

Point 4: Copyright license, Archives, Reuse: open CC0, with doi number, reusable.

Response 4: I have submited the dataset to a repository which assigned a doi, with a

 CC BY 4.0 licence.

“Dataset: https://doi.org/10.6084/m9.figshare.7857002.v1

Point 5: The authors declare having made a few corrections, without further describing the origin of these errors. A few more comments would be welcomed. The guess is that the new dataset shares the same level of quality than the previous one, and broken down into the same types of quality, excepting the time dimension.

Response 5: I have added this explanation to the section about the misplacements “MMQGIS uses Google maps as a base map, and it is advised to use a Google Earth background map to insure the good quality and precision of geocoding. In our case, since we have used our own shapefile as a background map, some districts were misplaced. Obvious misplacements, like the ones outside the land, were corrected automatically by dragging them into the nearest part of land. Other less noticeable ones, especially the ones over some communes’ boundaries, were handled individually by looking for their right commune, in some cases referring to Wikipedia, in others, to our knowledge of the area.

Point 6: the proposed formats are ok (CSV and ShapeFile). Would it be possible to provide a GeoJson as well?

Response 6: As I cannot add it in the published dataset, I have added it separately in a note at the end of the article

“An additional folder of the output maps in GeoJson format is available on this link
